# Evaluating the Effect of Tideglusib-Loaded Bioactive Glass Nanoparticles as a Potential Dentine Regenerative Material

**DOI:** 10.3390/ma15134567

**Published:** 2022-06-29

**Authors:** Akhil C. Rao, K. Vijay Venkatesh, Vidyashree Nandini, Dhanasekaran Sihivahanan, Ahmed Alamoudi, Hammam Ahmed Bahammam, Sarah Ahmed Bahammam, Bassam Zidane, Maha A. Bahammam, Hitesh Chohan, Nassreen H. Albar, Pradeep Kumar Yadalam, Shankargouda Patil

**Affiliations:** 1Department of Conservative Dentistry and Endodontics, SRM Dental College and Hospital, Kattankulathur, Kanchipuram, Chennai 603203, India; akhilcrao@gmail.com (A.C.R.); vijayvek@srmist.edu.in (K.V.V.); shivahad@srmist.edu.in (D.S.); 2Department of Prosthodontics and Implantology, SRM Dental College and Hospital, Kattankulathur, Kanchipuram, Chennai 603203, India; vidyashv@srmist.edu.in; 3Oral Biology Department, Faculty of Dentistry, King Abdulaziz University, Jeddah 80209, Saudi Arabia; ahmalamoudi@kau.edu.sa; 4Department of Pediatric Dentistry, College of Dentistry, King Abdulaziz University, Jeddah 80209, Saudi Arabia; habahammam@kau.edu.sa; 5Department of Pediatric Dentistry and Orthodontics, College of Dentistry, Taibah University, Medina 42353, Saudi Arabia; sarah.bahammam@gmail.com or; 6Restorative Dentistry Department, Faculty of Dentistry, King Abdulaziz University, Jeddah 21589, Saudi Arabia; bzidane@kau.edu.sa; 7Department of Periodontology, Faculty of Dentistry, King Abdulaziz University, Jeddah 80209, Saudi Arabia; mbahammam@kau.edu.sa; 8Executive Presidency of Academic Affairs, Saudi Commission for Health Specialties, Riyadh 11614, Saudi Arabia; 9Department of Restorative Dental Sciences, College of Dentistry, Jazan University, Jazan 45142, Saudi Arabia; drhiteshchohan@yahoo.co.in (H.C.); nalbar01@gmail.com (N.H.A.); 10Department of Periodontics, Saveetha Institute of Medical and Technical Sciences, Saveetha Dental College and Hospitals, Saveetha University, Chennai 602117, India; pradeepkumar.sdc@saveetha.com; 11Department of Maxillofacial Surgery and Diagnostic Sciences, Division of Oral Pathology, College of Dentistry, Jazan University, Jazan 45142, Saudi Arabia; 12Centre of Molecular Medicine and Diagnostics (COMManD), Saveetha Dental College & Hospitals, Saveetha Institute of Medical and Technical Sciences, Saveetha University, Chennai 600077, India

**Keywords:** tideglusib-BgNPs, dentine regeneration, tricalcium silicate-based cement

## Abstract

Dental pulp treatment is the least intrusive procedure currently available for preserving the vitality of the pulp. Several studies are underway to improve the bioactivity of pulp capping materials. Tideglusib isa potent anti-inflammatory, antioxidant, and a regenerative drug developed against Alzheimer’s disease and has been shown to be effective in the treatment of dental cavities. However, its bioactive properties encapsulated within the nanoparticles as a component of pulp capping material are largely unknown. In this study, tideglusib-loaded bioactive glass nanoparticles were synthesized (tideglusib-BgNPs) and mixed at various concentrations into the calcium silicate cement to testits physiomechanical and bioactivitiescompared with biodentine (control). The calcium silicate cement with 10wgt% tideglusib-BgNPs showed comparable physiomechanical properties to that of biodentine. Additionally, the assessment of cytotoxicity and bioactivity (cell proliferation, wound healing, and cell migration assays) showed increased bioactivity in terms of better wound healing, increased proliferation, and better migration of human dental pulp stem cells than biodentine. These findings suggest new opportunities to use tideglusib-BgNPs in pulp therapy.

## 1. Introduction

Pulp is a mesenchymal tissue located peripherally in close connection with the dentine matrix. These cells are termed odontoblasts, and they are specialized in the production of dentine. Pulp has a relatively abundant resource of stem cells that several tissue elements dynamically respond to via developmental, physiological (such as orthodontic or masticatory forces), or pathological stimuli. This allows the pulp to play an important role in tooth development [1]. The vitality of the pulp is maintained as it remains encased in a protective covering of dentine and enamel. Dental caries, mechanical factors, or trauma could cause the pulp to become exposed to the oral environment, which could lead to an infection. In the event of pulp exposure to the external environment, it could cause necrosis [2,3]. In such situations, a direct pulp capping procedure is performed to maintain the vitality of the pulp tissue. Pulp capping will stop serious damage or avoid more invasive and complicated procedures such as endodontic therapy or extraction [4].

Calcium hydroxide cement was extensively used in pulp capping procedures. Due to certain drawbacks, such as breakdown overtime and tunneled dentin bridges, researchers have now turned their attention to bioactive bioceramic materials [5]. The bioactivity of cement is described as a substance’s capacity to evoke a certain biological reaction in its contacted tissues [6]. For over a decade now, bioceramic cement materials such as mineral trioxide aggregate (MTA), bioceramic root repair material putty, and biodentine have gained popularity in restorative dentistry and endodontics [7]. Most of these materials show high biocompatibility, reduced toxicity, higher pH, radiopacity, and good sealing ability. Additionally, these materials have been shown to induce the proliferation and differentiation of dental pulpal stem cells [8]. Biodentine has drawn significant attention in dental applications. It is composed of tricalcium silicate, calcium carbonate, zirconium oxide, and calcium chloride and is used for apexification, retrograde fillings, root perforations, pulp capping, and dentine replacement. Recently, a more biological approach towards dentin regeneration was described by Neves et al. [9] who advocated the role of glycogen-synthase-kinase 3 (GSk-3).

Tideglusib is a thiadiazolidinone that inhibits neurodegeneration and inflammation [10]. In human neuroblastoma cells and murine primary neurons, it inhibits GSK-3 permanently, lowers tau phosphorylation, and avoids apoptosis. It stimulates stem cells and activates the Wnt/-cat signaling pathway for recovery. Presenting tideglusib at the site of tooth damage could activate this pathway, allowing stem cells to create reparative dentin and spontaneously restore the damaged dentin. Tideglusib is now being evaluated for use in a variety of neurodegenerative illnesses, most notably Alzheimer’s disease [11]. Moreover, compared to MTA, which uses a collagen sponge as a delivery method, tideglusib led to a lot more mineralization at the injury site [9].

Drug delivery systems are designed to simplify drug dosage and extend the duration of action, lowering drug-related side effects and maximizing the benefits of reduced frequency. In recent years, nanoparticles have become popular as carriers for controlled drug administration [12,13] because of their distinctive pore size, microporosities, increased surface area, excellent loading, and long-term release properties. It has been proposed that mesoporous nanoparticles can operate as transporters of drugs, assisting their loading and release [14]. With the development of their surface properties, characteristics, and evolution in the field of nanotechnology, bioactive glasses as drug delivery systems have found added application in the fields of medicine and dentistry. Their structures enable them to absorb drugs and facilitate sustainable release [15,16]. Mesoporous bioactive glass nanoparticles are evaluated as a delivery system with loaded antibodies, growth factors, and bone morphogenetic proteins [17,18,19]. Recent in vitro cytotoxicity and bioactivity studies using human dental pulp stem cells with bioactive glass nanoparticles showed increased proliferation and potential attachment of the stem cells [20]. However, tideglusib-loaded bioactive glass nanoparticles (tideglusib-BgNPs) have not been established for their application inregenerative dentistry. Further, sustained release of tideglusib from nanoparticles may contribute to enhancing the bioactivity of pulp capping material. The schematics workflow of this study is presented in Figure 1.

In this study, tideglusib-loaded bioactive glass nanoparticles (tideglusib-BgNPs) were synthesized. The physiomechanical properties (setting time, compressive strength, and alkalinity) and bioactivity on stem cells for pulp vitality were evaluated using a known concentration of tideglusib-BgNPs mixed with calcium silicate cement. The development of calcium silicate cement containing tideglusib-BgNPs will expand pulp therapy options.

## 2. Materials and Methods

The majority of the materials and chemicals, including the drug (tideglusib), were obtained from Sigma-Aldrich, Bangalore, India. All chemicals were used without further purification. For the preparation, only double distilled water was used as the solvent.

### 2.1. Nanoparticle Synthesis and Drug Loading

Mesoporous bioactive glass nanoparticles (BgNPs) were synthesized by the sol-gel method as described by Bae et al. [21]. Briefly, 2-ethoxyethanol, aqueous ammonia, calcium nitrate tetrahydrate, and ethanol were added to distilled water. Hexadecyltrimethylammonium bromide was then added to the mixture and agitated for 30 min at room temperature. Then tetraethyl orthosilicate was added by vigorous stirring for 4 h. The received white precipitate was ethanol washed and dried for 24 h at 60 °C. Finally, calcinations were performed at a temperature of 600 °C. Further, tideglusib (Sigma Aldrich, India) was loaded onto the nanoparticles by dissolving it in DMSO (2:1) and stirring continuously. A total of 2 mL of phosphate buffer (PBS) was added and agitated for 15 min. A total of 30 mg of bioactive glass nanoparticles were added and mixed overnight. Further, the loading efficiency was calculated by determining the concentration of tideglusib that remained in the supernatant after stirring using the UV-Spectrophotometer (LABINDIA UV 3000, Lab India Instruments Pvt. Ltd., Mumbai, India). The following formula was used to calculate the drug loading:Drug loading (%) = [((A − B))/A] × 100
where A and B represent the initial and final drug concentrations of the aqueous drug solution.

### 2.2. Characterization of Nanoparticles

The synthesized BgNPswith and without tideglusib were analyzed by X-ray Diffraction (XRD) and Fourier transform-infrared spectroscopy (FT-IR). X-ray diffraction patterns were obtained using a Seifert, JSO-DE BYEFLEX 2002, Ahrensberg, Germany. The functional group of the nanoparticles was investigated using the ATR method by an FT-IR spectrometer (PerkinElmer Inc., Waltham, MA, USA).

### 2.3. Preparation of Cement

Calcium silicate cement was prepared following the method described by Saravanapavan et al. containing tricalcium silicate, dicalcium silicate, and calcium oxide [22]. Briefly, 200 µL of nitric acid was added to 15 mL of distilled water. A total of 8.48 mL of TEOS was slowly added to this mixture. The solution was rapidly stirred for 1 h after the addition of 26.56 g of calcium silicate. Overnight drying of the mixture was performed at room temperature. The product was then heated to a temperature of 120 °C for 24 h and chelated at a temperature of 14,000 °C. After chelation, the material was ground and sieved through 52 µ mesh.

### 2.4. Classification of Cements intoExperimental Groups

The cement was mixed with the 5, 10, and 15 wgt% of BgNPs with and without tideglusib. The cement specimens were classified as group-A, group-B, and group-C. In group-A, cement specimens contain BgNPs without the drug, represented as X1 (5%), X2 (10%), and X3 (15%). Whereas group-B cement specimens contain tideglusib-loaded nanoparticles (tideglusib-BgNPs), indicated as D1 (5%), D2 (10%), and D3 (15%). For comparative analysis, group-C contains a plain biodentine without the nanoparticles.

### 2.5. Physiomechanical Evaluation

The physiomechanical properties such as setting time, compressive strength, and alkalinity were tested for cement specimens in groups-A and B and compared with group-C. Each specimen in a group was assessed in triplicate.

#### 2.5.1. Setting Time Assessment

The setting time was measured according to the ISO standards (ISO 6876:2012). The specimens were spread on a Teflon mold with a 1 mm thickness and a 10 mm diameter for the setting time analysis. Then the Gilmores apparatus, (Department of Dental Materials, Yenepoya University, Mangalore, India) with a 1/4 pound indenter, was vertically applied to the horizontal surface of the material for 5 s. The moment the indenter ceased to leave a distinct mark, it was recorded as the initial setting time.

#### 2.5.2. Evaluation of Compressive Strength

The compressive strength was measured according to ISO guidelines (ISO 9917-1:2003). The samples were prepared (6 mm in thickness and 4 mm in diameter) and allowed to set at 37 °C. The compressive strength was measured using a universal testing machine (INSTRON ELECTROPLUS E3000, Instron, Norwood, MA, USA) after 24 h with a 0.5 N load cell at a crosshead speed of 0.1 mm/min. The compressive strength was calculated using the formula C=4P/D2, where C is the compressive strength (MPa), P is the maximum load force before fracture (N), and D is the diameter (mm) of the specimen.

#### 2.5.3. Alkalinity Analysis

To test the alkalinity, the specimens were prepared (1-mm thickness and 5-mm diameter) and allowed to set completely. After setting, the specimens were inserted into 10 mL of deionized water. A pH meter (AP-1PLUS, Susima, Chennai, India) was used to measure the pH at 0, 1, 2, 3, and 24 h [23].

### 2.6. Drug Release

The release of the tideglusib from a specimen was tested by placing it in a vial containing 5 mL of PBS solution. For every 6, 12, 18, 24, 36, and 48 h, the amount of drug released in the solution was assessed. The evaluation was performed using a UV-Spectrophotometer (LABINDIA UV 3000, Lab India Instruments Pvt. Ltd., Mumbai, India), where the peak of 285 nm was calculated for tideglusib. Furthermore, the specimens were subjected to evaluation for biocompatibility.

### 2.7. Biocompatibility

Human dental pulp fibroblasts were obtained from the repository of Maratha Mandal’s NGH Institute of Dental Sciences and Research Centre, Belagavi, India. The fibroblasts were maintained in a 96-well microtiter plate containing DMEM supplemented with 10% heat-inactivated fetal calf serum (FCS), containing 5% of a mixture of gentamicin (10 µg), penicillin (100 units/mL), and streptomycin (100 µg/mL)in the presence of 5% CO_2_ at 37 °C for 48–72 h. Further, the following SETs were assessed for biocompatibility.

**SET-A**: Stem cells without the addition of specimen elute (negative control).

**SET-B**: Stem cells treated with biodentine elute.

**SET-C:** Stem cells treated with elute obtained from the cement specimen containing 10 wgt% tideglusib-BgNPs.

**SET-D:** Stem cells treated with elute obtained from the cement specimen containing 10 wgt% BgNPs (without tideglusib).

The elutes were prepared from each specimen (SET-B to D). For SET-B, the biodentine elute was prepared as per the manufacturers’ (Septodont, Saint-Maur-des-Fossés, France) instructions. Likewise, for SET-C and D, the cement specimens were first mixed with the distilled water (3:1) and made into cylinders using a Teflon mold with a dimension of 2 × 2 mm. Each cylindrical cement specimen was placed in the wells of sterile 96-well plates for 24 h at 37 °C for the material to set. Then, 1 mL of DMEM was added to each well and placed in the humidified incubator with 5% CO_2_ and 95% air at 37 °C for 24 h to obtain the extracts [24].

#### 2.7.1. MTT Test

The cell viability and cytotoxicity effect of the elutes were determined by the MTT assay as described by Mandrole et al. and Bubule et al. [25,26]. The cells were treated with various elutes, except SET-A. The elutes were incubated at 37 °C in a humid atmosphere containing 5% CO_2_ for 24 h. A stock solution of MTT was added (20 µL, 5 mg per ml in sterile PBS) to each well and incubated for four hours. The supernatant was then aspirated. The precipitated formalin-blue crystals were solubilized using 100 µL of DMSO. The optical density was measured at a wavelength of 595 nm by using LISA plus. The results represent an average of triplicates. The concentration based on the optical density (OD) of elute-treated cells was reduced by 50% with respect to the negative control.
Cell survival (%) = (mean OD of test elute)/(mean OD of negative control) × 100

#### 2.7.2. Analysis of Scratch Wound Healing Assay

Stem cells were seeded on 96-well cell culture plates with a growth medium. After 24 h of culture, a scratch wound was created in the middle of the cell layer using a sterile pipette. After scratching, the debris was rinsed off with PBS. The healing process was permitted to continue in the absence (control) and presence of elutes (elutes prepared for biocompatibility) after 24 h of culture. Using a phase contrast microscope, the “wound” was observed at 0, 3, and 24 h. An image was used to determine the wound healing area. Each group was evaluated in quadruplicate.

#### 2.7.3. Trans-Well Migration Assay

In a biosafety hood, the cells were detached from the tissue culture plate using a 0.25% Trypsin EDTA solution. Then, the cells were pelleted by centrifugation, and the supernatant was aspirated. The cells were re-suspended in serum-free cell culture media containing 0.1% bovine serum albumin (BSA). For this experiment, 1000 µL (106 cells/mL) of cell solution was placed on top of the filter membrane in a trans-well (8 µm pore size) insert and incubated for 10 min at 37 °C with 5% CO_2_ to allow the cells to settle down. Using a pipette, very carefully, 3 mL of media containing FBS was added for negative control, and then the elutes were added to the bottom of the lower chamber of a 6-well plate. The plate was incubated for one day (24 h). After 24 h, the insert was removed. Excess media was removed with a cotton-tipped applicator without damaging the membrane. Then, 4 to 5 mL of 70% ethanol was added to each well. The insert was then placed in the well containing ethanol and incubated for 10 min to allow for fixation. The insert was removed from the well. Excess ethanol was removed with a cotton applicator and allowed to dry for 10–15 min. Crystal violet at a concentration of 0.2% was added to a 6-well plate, and the insert was placed into the well. The insert was left in the crystal violet stain for 5 to 10 min at room temperature. The crystal violet stain was gently removed from the membrane with a cotton-tipped applicator. Then the insert was dipped in distilled water until the stain was removed. The trans-well membrane was allowed to dry. Stained cells were counted under an inverted microscope in different fields to get an average sum of cells that had migrated and attached to the underside of the membrane.

### 2.8. Statistical Evaluation

GraphPad prism5 software (GraphPad Software Inc.; San Diego, CA, USA) was used for statistical analysis. The collected quantitative data were expressed as mean ± standard deviation (SD). An analysis of variance (ANOVA) was performed to determine the significance between the experimental groups by adopting the Tukey post-hoc test. A *p*-value of 0.05 is considered for statistical significance.

## 3. Results

### 3.1. Microstructural Examination

The characteristics of the synthesized nanoparticles were tested using a variety of analytical instruments. For BgNPs, the FT-IR spectra showed the asymmetric stretching mode of Si-O-Si produced a peak roughly at 1030 cm^−1^(Figure 2A).

At 558 and 788 cm^−1^, Si-O-Si asymmetric stretching and vibration were recorded. Water absorption of OH on its surface occurred at 1485 cm^−1^, and OH vibrations were noticed at 2925 cm^−1^ [21]. Likewise, in XRD, the peak was dispersive, and there was no sharp peak of diffraction, which demonstrates that the nanoparticles were typically amorphous. Particularly at 2theta, around 24 (degrees), a broad peak was noticed, which represents the typical amorphous structure characteristic of silicate glass as described by Chen et al. [27] (Figure 3).

Further, the FT-IR analysis of tideglusib-BgNPs showed peaks between 1725 and 1562 cm^−1^. These peaks indicate the presence of tideglusib based on carbonyl-C=O and amide groups (Figure 2B) [28].

### 3.2. Drug Encapsulation

The tideglusib loading capacity of the BgNPs was evaluated following the drug loading procedure as described in the Section 2.1. The average encapsulating efficiency of nanoparticles for tideglusib was found to be 62.33 ± 2.0% (Figure 4).

### 3.3. Physiomechanical Properties and Drug Release

All synthesized cement specimens were tested for their physiomechanical efficiencies based on their (1) setting time, (2) compressive strength, and (3) alkalinity. A significant difference was noticed in setting time, compressive strength, and alkalinity between the weights (5, 10, and 15 wgt%) across the specimen groups (group-A and group-B) when compared to the control (group-C: biodentine). In particular, the specimens D2 prepared with 10wgt% tideglusib-BgNPs in group-B showed equal or better performance in most of their physiomechanical tests than biodentine (*p* > 0.05). For instance, the D2 specimen takes less or equal setting time, and compressive strength with no statistical difference (*p* > 0.05) compared to standard biodentine (Figure 5A,B).

Further, the alkalinity properties of D2 based on timeintervalswere noticed to be comparably equal or less to those of biodentine (Figure 6).

Although X2 (group-A) showed similar properties to D2, our prime interest was to test the tideglusib-BgNPs specimen for the subsequent analysis. Thereby, the elute from X2 (group-A) was used as part of the control (SET-D) for the biocompatibility assessment. As a result of the physiomechanical assessment, the drug release from the D2 specimen was tested in a PBS solution at 6, 12, 18, 24, 36, and 48 h. A sustained release of the drug was noticed between 18 and 48 h of the assessment time period (Figure 7).

### 3.4. Biocompatibility

The biocompatibility was tested using the elute from each specimen (SET-B–D). The MTT assay (Figure 8A) showed a significant difference between the experimental specimens (SET-B–D) compared to the control (SET-A). In particular, the SET-C elute showed 90% cell viability, whereas SET-B (Biodentine elute treated stem cells) showed limited viability of 64% (*p* < 0.05) (Figure 8A). SET-D, on the other hand, had a cell survival rate of around 30%. (Figure 8A). The presence of nitric acid, which was used in the material’s synthesis, may contribute to a lower cell survival rate. Although the same material was used in SET-C, the presence of the medication (tideglusib) within BgNPs protected the cells from toxicity. However, the tideglusib-BgNPs specimen (SET-C) outperformed the commonly used biodentine in terms of cytotoxicity assessment (SET-B).

Similarly, better healing or reduction in the scratch area was noticed at 24 h with SET-C elutein the scratch wound healing assay (Figure 8B). This may be due to the migration ability of cells to the scratched area. Further, the trans-well-based migration analysis at the end of 24 h revealed maximum migration of the cells in SET-C when compared to SET-B. Whereas SET-D without tideglusib showed no migration of the cells (Figure 8C) compared to other SETs, including SET-A. These findings clearly show that tideglusib-loaded BgNPs play a critical role in cell migration and scratch wound healing properties.

## 4. Discussion

This study aimed to develop calcium silicate cement containing tideglusib-loaded bioactive glass nanoparticles as a direct pulp-capping material. Tideglusib is involved in inhibiting GSK-3; they were loaded into bioactive-glass nanoparticles and mixed with calcium silicate cement, which acts as a base to seal off the pulp tissue. We synthesized the mesoporous bioactive glass nanoparticles using the sol-gel technique [21], then we loaded them with tideglusib and characterized them using multiple analytical techniques. For instance, we confirmed the nanoparticle’s characteristics using FTIR and XRD. Further, the prepared nanoparticles were mixed with the calcium silicate cement specimen at various concentrations and were tested for physiomechanical parameters, such as setting time, compressive strength, and alkalinity [29]. Among them, the cement specimens with 10wt% tideglusib-BgNPs were outperformed based on the physiomechanical properties and met the properties of standard commercial biodentine. For instance, the compressive strength and alkalinity properties were noticeably equal to those of biodentine. Biodentine is relatively easier to manipulate and has higher compressive and flexural strength, high biocompatibility, and excellent bioactivity, making it a proven pulp capping and root end closure material. Interestingly, the specimens prepared with 10 wgt% showed better performance than biodentine regarding setting time, whereas no appreciable results were noticed with other specimens prepared at 5 or 15 wgt%. These findings suggest that tideglusib-BgNPs play a critical role in regulating the physiomechanical properties of cement.

In general, the nanoparticles act as a potential drug delivery system, which helps to improve the bioavailability of the drug by maintaining optimal concentration within the therapeutic range, reducing side effects [30]. Herein, mesoporous BgNPs were used as a drug-delivery system. The porous structure of the BgNPs facilitated the acceptance of the tideglusib into its micropores via capillary action [21]. The nano-sized drug delivery system was introduced for treating chronic diseases for long-term sustained delivery of drugs. Further, it is desirable for the drug delivery system to have a higher drug loading capacity that reduces the number of delivery vehicles [31]. Hostiuc et al. [32], in 2019, raised concern about the systemic action of tideglusib and its introduction to endodontics and restorative procedures. Intra-dental application of tideglusib could cause increased release time, which could potentiate a significant local effect. In such situations, a nano-based drug delivery system could prove beneficial since the dosage and release could be controlled. Furthermore, the more of the surface of the nanoparticles is exposed, the higher their chances of interacting with the surrounding biomolecules. This raises the possibility of toxicity in nanoparticles with smaller sizes [33].The assessment of our nanoparticles based on the loading procedure employed facilitates minimal drug loading (62%), which was beneficial from an ethical point of view. In the drug release analysis, the maximum release was observed in the initial 6–18 h followed by sustained release with tapering doses up to 48 h. Due to controlled release, the dosage can be regulated to obtain a better result with minimal side effects [34].

In recent times, the use of numerous direct pulp-capping materials, including biodentine, has become standard procedure. In order for these materials to work, they need to be in contact with the biological tissue. Additionally, they should be able to seal the defect off from the wet oral environment, as this is one of the important features that are being considered [6]. Luo et al. [35] investigated the effect of biodentine on human dental pulp stem cells and discovered that when positioned in contact with the pulp, biodentine greatly enhances stem cell proliferation, migration, and adhesion. Additionally, it has been observed that biodentine causes irritation in the exposed area, leading to coagulation necrosis; this results in precursor cell division and migration to the substrate surface, as well as precursor cell addition and cytodifferentiation into odontoblast-like cells [36,37]. Thus, we demonstrated the effectiveness of the cement specimens through their elutes by testing for their cytotoxicity (MTT assay), wound healing, and cell migration. It is noteworthy that tideglusib is a thiadiazolidinone that prevents inflammation and neurodegeneration. Thereby, the elute from a cement specimen with tideglusib-BgNPs(SET-C) was outperformed, showing immediate localization of cells in the scratch area compared to the biodentine even in the early stages of the investigation. By 24 h, the scratch had completely healed by treating it with SET-C. The results were justified with the cytotoxicity assay (MTT assay) and trans-well migration assay. Interestingly, the SET-C showed 90% cell viability, whereas biodentine showed only 64% cell viability. The migration assay was used to determine cells’ capacity to respond directionally to various chemoattractants, including chemokines, growth factors, lipids, and nucleotides. By the end of 24 h, maximum cell localization was seen for the elute from SET-C. Surprisingly, there was no cell migration seen in SET-D. This implies that tideglusib-loaded BgNPs are the principal driver of biological action. Tideglusib inhibits glycogen-synthase-kinase 3 (GSK-3) that regulates the Wnt/-cat signaling pathway, and activates the cellular-based repair mechanism. According to the observations of Nair et al., it takes around a week for a pulp-capped tooth to become free of inflammation [38]. The anti-inflammatory properties of tideglusib could be beneficial in inducing a faster healing response. However, more research is needed to figure out how tideglusib works as an anti-inflammatory to produce a healing effect.

On the other hand, tideglusib is being studied for its potential use in Alzheimer’s disease. During the phase two clinical trials, the medication was well tolerated at doses of up to 1200 mg/kg for a period of 14 days with no side effects [11]. The amount of medication that was released from our specimen was less than the critical dose. To our knowledge, this is the first study demonstrating the effect of tideglusib-loaded bioactive glass nanoparticles (tideglusib-BgNPs) in the dentistry application of pulp capping. Further, our study provides time-dependent sustained release of tideglusib from BgNPs within the cement and adds evidence towards the enhancement of physicochemical and bioactivity influenced by tideglusib-BgNPs in the capping cement. Nonetheless, there are a few limitations that need to be addressed: (1) Systemic action of the medicine must be determined and recorded, even in the smallest dose; (2) The effect of tideglusib-BgNPs at the location of TGF-B1 and alkaline phosphatase, as well as its involvement in the induction of inflammation, needs to be evaluated; (3) In the drug release experiment (Figure 7), sustained drug release was noticed after 18 h. As a result, the majority of biological evaluations were performed after 24 h; multiple-point evaluations before and after 24 h would be appropriate. However, within the confines of this investigation, we discovered that the tideglusib bioactive glass nanoparticles have the ability to enhance the bioactivity of the cement. It is possible that the prolonged release of the medication at the site of interest is responsible for the significant contribution that tideglusib made to bioactivity. This shows that the tideglusib-bioactive glass nanoparticles could be a potential component with in the material in the field of regenerating dentine.

## 5. Conclusions

In this study, we synthesized BgNPs that could encapsulate significant amounts of tideglusib. When compared to the commercial alternative (biodentine) for dental pluping, we demonstrated that the tideglusib-BgNPs specimen (tideglusib-BgNPs within the calcium silicate cement) had enhanced the physiomechanical properties. The findings revealed that tideglusib-BgNPs specimens were cytocompatible and did not cause significant toxicity, implying that they are safe for biomedical use. Furthermore, when compared to biodentine, tideglusib-BgNPs specimens showed increased bioactivity in terms of wound healing, proliferation, andmigration of human dental pulp stem cells. Despite the fact that calcium silicate cement (without tideglusib-BgNPs) was found to be cytotoxic, adding tideglusib-BgNPs to the cement overcomes the toxicity and increases cell proliferation and migration, allowing for better treatment. Overall, our results suggest that tideglusib-BgNPs could be used in pulp therapy, which requires faster dentin formation and, as a result, faster healing and repair of the dentin-pulp complex. This opens up new possibilities for regenerative endodontics.

## Figures and Tables

**Figure 1 materials-15-04567-f001:**
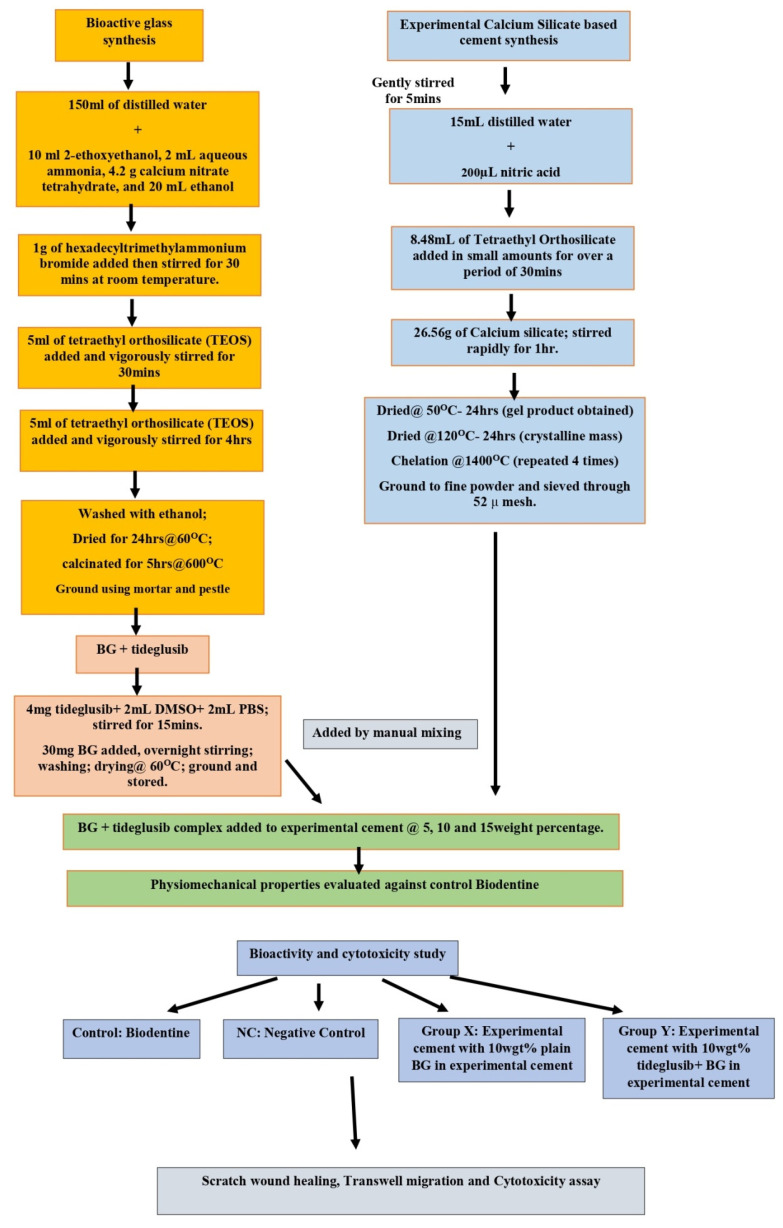
Schematic workflow of this study.

**Figure 2 materials-15-04567-f002:**
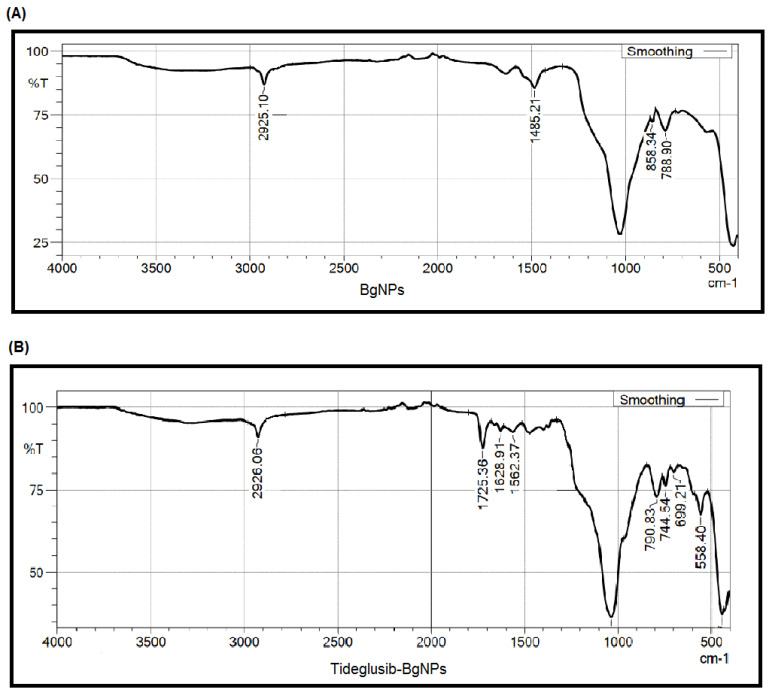
Microstructural behavior of the nanoparticles established using FT-IR (**A**) and XRD (**B**).

**Figure 3 materials-15-04567-f003:**
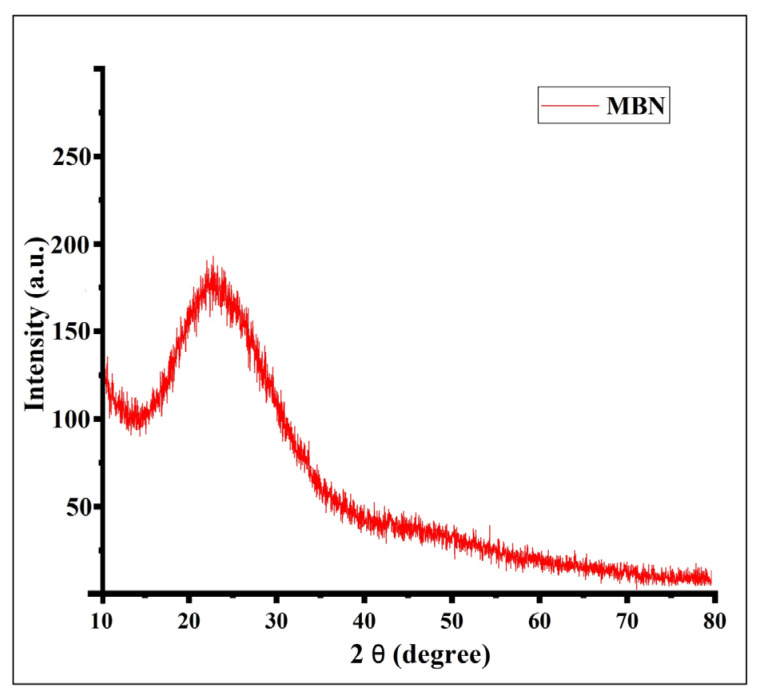
FT-IR spectrum presenting the functional group for the nanoparticles loaded with the tideglusib.

**Figure 4 materials-15-04567-f004:**
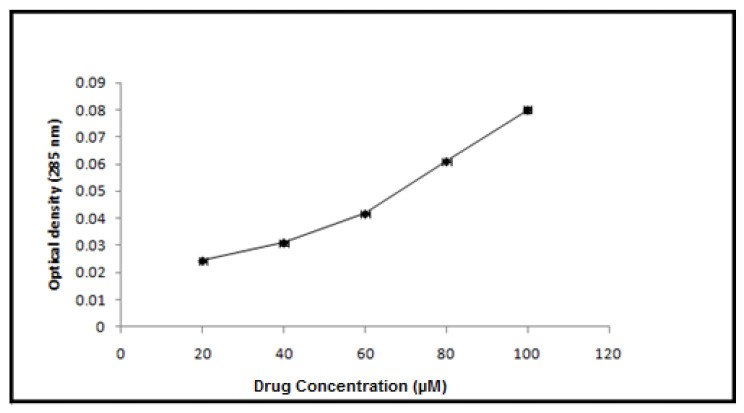
A line graph depicting the efficiency with which nanoparticles encapsulate tideglusib.

**Figure 5 materials-15-04567-f005:**
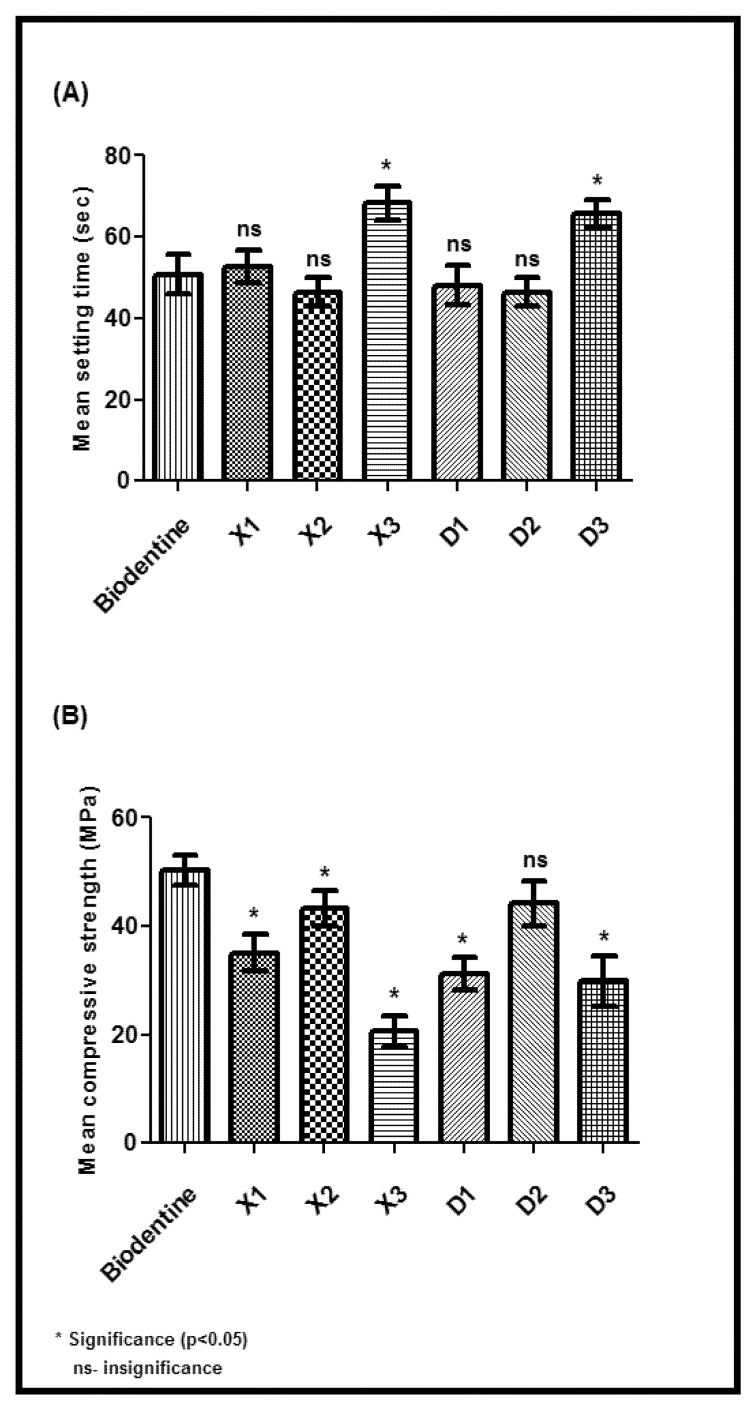
Physiomechanical properties; (**A**) setting time and (**B**) compression strength test.

**Figure 6 materials-15-04567-f006:**
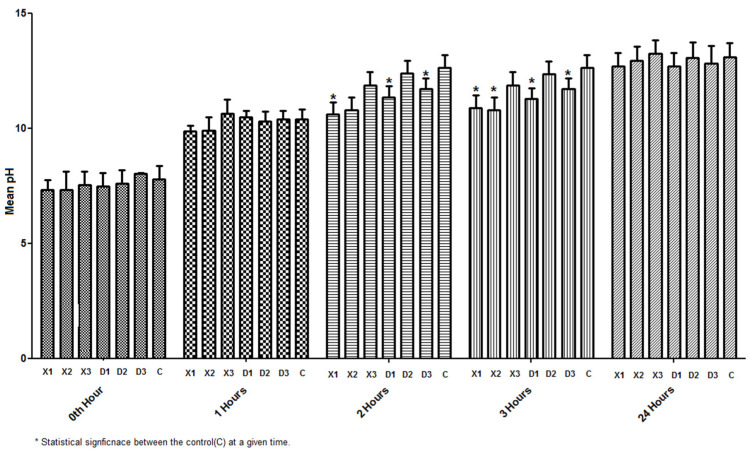
Physiomechanical property assessment based on alkalinity from 0 to 24 h.

**Figure 7 materials-15-04567-f007:**
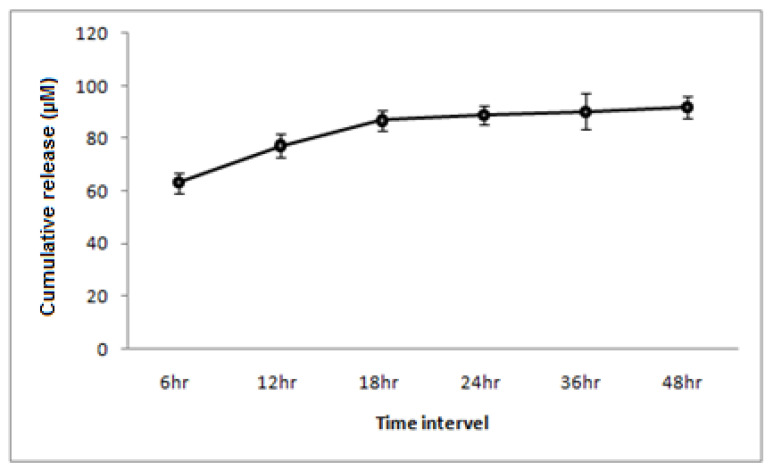
Line diagram for drug release efficiency from the specimen (10%wtg) from 6 to 48 h.

**Figure 8 materials-15-04567-f008:**
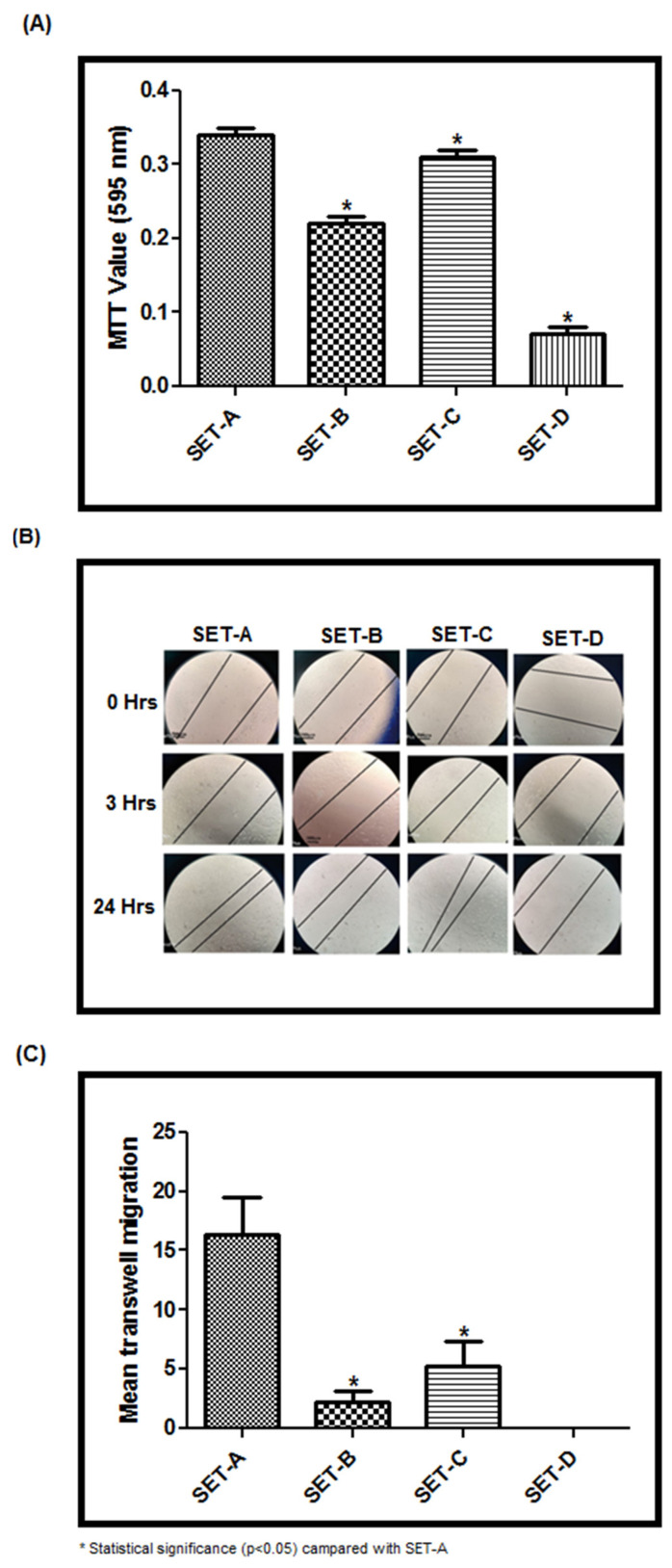
(**A**) MTT assay; (**B**) Scratch wound healing assay; (**C**) Cell migration assay to test biocompatibility.

## Data Availability

Not applicable.

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
