# Peer review of "Evaluating the Effect of Tideglusib-Loaded Bioactive Glass Nanoparticles as a Potential Dentine Regenerative Material"

_materials, 2022, doi:10.3390/ma15134567_

Round 1
Reviewer 1 Report
This article was a revision of the previous manuscript, and this reviewer appreciates the authors’ careful responses to my previous comments. The quality of the manuscript does have improved significantly. However, the major concerns on the differences between TEM and SEM images before and after drug loading are still unclear. Below are some detailed comments.
- This reviewer suggests the authors consider taking the SEM and TEM images out of the manuscript. FTIR results are already sufficient to illustrate the success of drug loading and additional TEM images didn’t offer new information. Please understand that, from the reviewer’s perspective, it is better to keep the unclear observations unpublished, rather than publish results that are misleading.
- In the response, the authors mentioned that EELS and EDS mapping are not available due to a lack of facilities. This is understandable but this cannot be an acceptable way to resolve the existing problems in the manuscript. This further suggests that all the statements and conclusions drawn from electron micrographs should have been removed.
Author Response
REVIEWER 1:
Comment 1. This reviewer suggests the authors consider taking the SEM and TEM images out of the manuscript. FTIR results are already sufficient to illustrate the success of drug loading and additional TEM images didn’t offer new information. Please understand that, from the reviewer’s perspective, it is better to keep the unclear observations unpublished, rather than publish results that are misleading.
Response: As suggested by the reviewer we have removed the SEM and TEM data in the manuscript.
Comment 2. In the response, the authors mentioned that EELS and EDS mapping are not available due to a lack of facilities. This is understandable but this cannot be an acceptable way to resolve the existing problems in the manuscript. This further suggests that all the statements and conclusions drawn from electron micrographs should have been removed.
Response: As suggested by the reviewer we have removed the electron micrographs data in the manuscript.

Reviewer 2 Report
The manuscript entitled „Evaluating the effect of experimental calcium silicate-based cement with bioactive glass-tideglusib nanoparticles as a potential dentine regenerative material“ is in an interesting field of research. However, the current presentation of the manuscript is not on an appropriate level for publication in MDPI Materials. I hope the comments and suggestions below will help the authors to increase the quality of the manuscript.
Overall, the manuscript needs English grammar and spelling checks.
Abstract:
From the abstract and title, it is not clear did the authors used both calcium silicates and bioactive glass. Which calcium silicate?
Introduction
The bracket for references should be [], and not ().
The authors should highlight the scientific impact of this study. Did other researchers try to do something similar or the same? Highlight the novelty.
Materials and methods
The bioactive nanoparticles and calcium silicate-based ceramics (still not defined which one) were obtained according to another study. Authors should think about how to improve from point of novelty of this study, as the used materials were already published.
The quality of Fig. 1. is too low and it cannot be properly read.
Results:
XRD results in Fig. 2 are not properly explained.
Overall figure presentation is not appropriate for publication.
Figures 7. and 8. Statistical calculations are required to determine a significant difference. Statistical calculations are missing on all graphs with error bars.
In the manuscript, it is not defined which calcium silicate cement was obtained and used. Is it wollastonite, larnite, dicalcium silicate, etc.?
The manuscript is not presented at the required level. There is a lack of novelty and scientific impact as used materials are The manuscript entitled „Evaluating the effect of experimental calcium silicate-based cement with bioactive glass-tideglusib nanoparticles as a potential dentine regenerative material“ is in an interesting field of research. However, the current presentation of the manuscript is not on an appropriate level for publication in MDPI Materials. I hope the comments and suggestions below will help the authors to increase the quality of the manuscript.
Overall, the manuscript needs English grammar and spelling checks.
Abstract:
From the abstract and title, it is not clear did the authors used calcium silicates and bioactive glass. Which calcium silicate?
Introduction
The bracket for references should be [], and not ().
The authors should highlight the scientific impact of this study. Did other researchers try to do something similar or the same? Highlight the novelty.
Materials and methods
So the bioactive nanoparticles and calcium silicate-based ceramics (still not defined which one) were obtained according to another study. Authors should think about how to improve from point of novelty of this study, as the used materials were already published.
The quality of Fig. 1. is too low and it cannot be properly read by the reader.
Results:
XRD results in Fig. 2 are not properly explained.
Figure presentation is not appropriate for high-quality journals as the Materials MDPI.
Figures 7. and 8. Statistical calculations are required to determine a significant difference.
Statistical calculations are missing on all graphs with error bars.
Still in the manuscript, it is not defined which calcium silicate cement was obtained and used. I sit wollastonite, larnite, dicalcium silicate, etc.?
The manuscript is not presented at the required level. There is a lack of novelty and scientific impact as two used materials are previously published.
Author Response
Reviewer 2:
Comment 1. Overall, the manuscript needs English grammar and spelling checks.
Response: the manuscript is now proof read by the English language editor . The current version of the manuscript is be free from the spelling and grammatical errors.
Abstract:
Comment 1: From the abstract and title, it is not clear did the authors used both calcium silicates and bioactive glass. Which calcium silicate?
Response: The title and abstract is now corrected. We prepared the calcium silicate cement by following the Saravanapavan et al., 2003 method. Simultaneously, the tideglusib loaded bioactive glass nanoparticles (tideglusib-BgNPs) was prepared using Bae et al., 2019 method. Further, the known concentration of tideglusib-BgNPs was then mixed to the calcium silicate cement to evaluate the mechanical and biological properties.
Reference:
Saravanapavan P, Hench LL. Mesoporous calcium silicate glasses. I. Synthesis. J Non-CrystSolids. 2003 Apr;318(1–2):1–13.
Bae J, Son WS, Yoo KH, Yoon SY, Bae MK, Lee DJ, et al. Effects of Poly(Amidoamine) Dendrimer-Coated Mesoporous Bioactive Glass Nanoparticles on Dentin Remineralization. Nanomaterials. 2019 Apr 10;9(4):591.
=========================================================
Introduction
Comment 2: The bracket for references should be [], and not ().
Response: Suggested changes were made throughout the manuscript.
Comment 3: The authors should highlight the scientific impact of this study. Did other researchers try to do something similar or the same? Highlight the novelty.
Response: Novelty and any similar studies were described in the introduction section.
============================================================================
Materials and methods
Comment 4: The bioactive nanoparticles and calcium silicate-based ceramics (still not defined which one) were obtained according to another study.
Response: We have clearly mentioned in the manuscript. Please see methodology section "2.1. Nanoparticle synthesis and drug loading" for the preparation (based on Bae et al., 2019 ) and loading of the tideglusib into the bioactive glass nanoparticles (tideglusib-BgNPs). Also, please refer methodology section "2.3. Preparation of cement" for the cement preparation by following Saravanapavan et al., 2003 method. The tideglusib-BgNPs was then mixed to the calcium silicate cement to evaluate the mechanical and biological properties for dental application.
Reference:
Saravanapavan P, Hench LL. Mesoporous calcium silicate glasses. I. Synthesis. J Non-CrystSolids. 2003 Apr;318(1–2):1–13.
Bae J, Son WS, Yoo KH, Yoon SY, Bae MK, Lee DJ, et al. Effects of Poly(Amidoamine) Dendrimer-Coated Mesoporous Bioactive Glass Nanoparticles on Dentin Remineralization. Nanomaterials. 2019 Apr 10;9(4):591.
================================================================================
Comment 5: Authors should think about how to improve from point of novelty of this study, as the used materials were already published.
Response: Thanks for the reviewer suggestion. Although, our study uses the previously reported cement and nanoparticles, the novelty of the study is based on two-fold 1) utility of tideglusib encapsulated nanoparticles in dentistry application. 2) Demonstrating the physiomechanical and bioactivity of the cement with the tideglusib-BgNPs on stem cells. The above novelty statements were added to the revised manuscript (abstract, introduction and discussion section).
Comment 6: The quality of Fig. 1. is too low and it cannot be properly read.
Response: We now provided the quality figure (Fig1).
===============================================================================
Results:
Comment 7: XRD results in Fig. 2 are not properly explained.
Response: the XRD result is now explained better.
Comment 8: Overall figure presentation is not appropriate for publication.
Response: we reworked on the figures for better quality.
Comment 8: Figures 7. and 8. Statistical calculations are required to determine a significant difference.
Response: Statistical analysis was performed and to determine the significance.
Comment 9: Statistical calculations are missing on all graphs with error bars.
Response: Figures were now better presented indicating the statistical significance
Comment 10: In the manuscript, it is not defined which calcium silicate cement was obtained and used. Is it wollastonite, larnite, dicalcium silicate, etc.?
Response: Thanks for the suggestion, please refer methodology section "2.3. Preparation of cement" for the cement preparation by following Saravanapavan et al., 2003 method. The calcium silicate cement was prepared containing tricalcium silicate, dicalcium silicate and calcium oxide by following the method described by Saravanapavan et al.,2003.
Reference:
Saravanapavan P, Hench LL. Mesoporous calcium silicate glasses. I. Synthesis. J Non-CrystSolids. 2003 Apr;318(1–2):1–13.

Round 2
Reviewer 1 Report
The authors have addressed all my concerns and I don't have additional comments.
Author Response
Reviewer 1:
Comment: The authors have addressed all my concerns and I don't have additional comments.
Response: Thank you for taking the time to read our response. All of the proposed comments has considerably improved the manuscript's quality.
Reviewer 2:
Comment 1: The authors improved the submitted manuscript regarding abstract, title and introduction issues. The manuscript needs to be checked for typos as there are a significant number of spelling and grammar errors.
Response: Thanks for the suggestion, we corrected the spelling and grammatical issues.
Comment 2: The author's answer to the comment regarding the novelty of the study was: 2) Demonstrating the physiomechanical and bioactivity of the cement with the tideglusib-BgNPs on stem cells.“
Response: Thank you for accepting our feedback on the manuscript's novelty.
Comment 3: According to my observations, MTT assay after 24 h has shown a negative effect (cytotoxicity) for SET D samples (Stem cells treated with elute obtained from the cement specimen containing 10wgt% BgNPs (without tideglusib). According to my calculations from the graph (I only could read approximate numbers from the graph), the cell survival is around 27 % (in comparison to control SET A). In cell culture, cell survival below 70% suggests the cytotoxicity of a material.
Response: Regarding cytotoxicity in SET-D, we acknowledge the reviewer's observation. In SET-D, the cell survival rate was around 30%. The presence of nitric acid during the material production may have resulted in a reduction in cell survival percentage. Although, same material was used in the SET-C, the presence of the medication (tideglusib) protected the cells against toxicity. It should be highlighted that the drug-loaded cement outperformed the commonly used biodentine (SET-B).
Comment 4: The authors did not even mention in describing the results SET D.
Response: We've incorporated the information from the previous comment (Comment:3) on the behaviour of SET-D in comparison to SET-A, as advised.
Comment 5 :: A similar problem is evident in the analysis of cell migration assay. Results are not compared to the control (SET A, negative control).
Response: We were unable to do statistical analysis between (SET-A vs.SET-D) with null value since no (zero) cell migration was observed in SET-D. The revised manuscript, however, now includes a comparison of SET-A and SET-D behaviour. We agree with the reviewer, and our findings clearly show that drug play a critical role in cell migration. Although the presence of a medication within the nanoparticles aids in the maintenance of sustain release.
Comment 6: Biological characterization results suggest that obtained materials are not appropriate for application in tissue engineering.
Response: Despite the fact that BgNPs (without tideglusib) did not perform well in biological characterization, the BgNPs with medication outperformed than the current existing Biodentine material. Hence BgNPs with medication will be appropriate for dental application (primary objective of this study).
Comment 7: Another drawback of obtained biological characterization is that all experiments were done after 24 hours. Biological characterization needs to have at least two-time points, and evaluation after only 24 hours is not appropriate.
Response: In drug release experment ( figure. 7) we noticed constant release of drug after 18hrs. Hence, most of the biological characterization were after 24 hr. However, we inculde the reviewer statement as the study limitation.
Comment 8: The conclusion is not adequately written.
Response: The conclusion has been rewritten to include a message in response to the aforementioned comments.
Comment 9:: The quality of the images and presentation is still low and not appropriate for publication in the journal MDPI Materials (Q1/Q2, IF = 3.623, 2020).
Response: Despite the fact that we have separate image files with higher quality, we follow the procedure by adding (inserting) all figures into the MDPI template document. During the manuscript production process, however, we will provide individual TIF image file with the required quality.

Reviewer 2 Report
The authors improved the submitted manuscript regarding abstract, title and introduction issues. The manuscript needs to be checked for typos as there are a significant number of spelling and grammar errors.
The author's answer to the comment regarding the novelty of the study was:
„2) Demonstrating the physiomechanical and bioactivity of the cement with the tideglusib-BgNPs on stem cells.“
According to my observations, MTT assay after 24 h has shown a negative effect (cytotoxicity) for SET D samples (Stem cells treated with elute obtained from the cement specimen containing 10wgt% BgNPs (without tideglusib). According to my calculations from the graph (I only could read approximate numbers from the graph), the cell survival is around 27 % (in comparison to control SET A). In cell culture, cell survival below 70% suggests the cytotoxicity of a material. The authors did not even mention in describing the results SET D.
A similar problem is evident in the analysis of cell migration assay. Results are not compared to the control (SET A, negative control).
Biological characterization results suggest that obtained materials are not appropriate for application in tissue engineering. Another drawback of obtained biological characterization is that all experiments were done after 24 hours. Biological characterization needs to have at least two-time points, and evaluation after only 24 hours is not appropriate.
The conclusion is not adequately written.
The quality of the images and presentation is still low and not appropriate for publication in the journal MDPI Materials (Q1/Q2, IF = 3.623, 2020).
Author Response
Reviewer 1:
Comment: The authors have addressed all my concerns and I don't have additional comments.
Response: Thank you for taking the time to read our response. All of the proposed comments has considerably improved the manuscript's quality.
Reviewer 2:
Comment 1: The authors improved the submitted manuscript regarding abstract, title and introduction issues. The manuscript needs to be checked for typos as there are a significant number of spelling and grammar errors.
Response: Thanks for the suggestion, we corrected the spelling and grammatical issues.
Comment 2: The author's answer to the comment regarding the novelty of the study was: 2) Demonstrating the physiomechanical and bioactivity of the cement with the tideglusib-BgNPs on stem cells.“
Response: Thank you for accepting our feedback on the manuscript's novelty.
Comment 3: According to my observations, MTT assay after 24 h has shown a negative effect (cytotoxicity) for SET D samples (Stem cells treated with elute obtained from the cement specimen containing 10wgt% BgNPs (without tideglusib). According to my calculations from the graph (I only could read approximate numbers from the graph), the cell survival is around 27 % (in comparison to control SET A). In cell culture, cell survival below 70% suggests the cytotoxicity of a material.
Response: Regarding cytotoxicity in SET-D, we acknowledge the reviewer's observation. In SET-D, the cell survival rate was around 30%. The presence of nitric acid during the material production may have resulted in a reduction in cell survival percentage. Although, same material was used in the SET-C, the presence of the medication (tideglusib) protected the cells against toxicity. It should be highlighted that the drug-loaded cement outperformed the commonly used biodentine (SET-B).
Comment 4: The authors did not even mention in describing the results SET D.
Response: We've incorporated the information from the previous comment (Comment:3) on the behaviour of SET-D in comparison to SET-A, as advised.
Comment 5 :: A similar problem is evident in the analysis of cell migration assay. Results are not compared to the control (SET A, negative control).
Response: We were unable to do statistical analysis between (SET-A vs.SET-D) with null value since no (zero) cell migration was observed in SET-D. The revised manuscript, however, now includes a comparison of SET-A and SET-D behaviour. We agree with the reviewer, and our findings clearly show that drug play a critical role in cell migration. Although the presence of a medication within the nanoparticles aids in the maintenance of sustain release.
Comment 6: Biological characterization results suggest that obtained materials are not appropriate for application in tissue engineering.
Response: Despite the fact that BgNPs (without tideglusib) did not perform well in biological characterization, the BgNPs with medication outperformed than the current existing Biodentine material. Hence BgNPs with medication will be appropriate for dental application (primary objective of this study).
Comment 7: Another drawback of obtained biological characterization is that all experiments were done after 24 hours. Biological characterization needs to have at least two-time points, and evaluation after only 24 hours is not appropriate.
Response: In drug release experment ( figure. 7) we noticed constant release of drug after 18hrs. Hence, most of the biological characterization were after 24 hr. However, we inculde the reviewer statement as the study limitation.
Comment 8: The conclusion is not adequately written.
Response: The conclusion has been rewritten to include a message in response to the aforementioned comments.
Comment 9:: The quality of the images and presentation is still low and not appropriate for publication in the journal MDPI Materials (Q1/Q2, IF = 3.623, 2020).
Response: Despite the fact that we have separate image files with higher quality, we follow the procedure by adding (inserting) all figures into the MDPI template document. During the manuscript production process, however, we will provide individual TIF image file with the required quality.

This manuscript is a resubmission of an earlier submission. The following is a list of the peer review reports and author responses from that submission.
Round 1
Reviewer 1 Report
In this work, the authors reported a method to combine tideglusib, mesoporous nanoparticles, and silicate-based cement for repairment of pulp tissue. The results are interesting and potentially useful. However, the results included in the manuscript have some critical errors and misleading statements that need to be addressed before the manuscript can be accepted.
Major comments:
- The abstract needs to be further polished and detailed experimental results should not be included in the abstract. The abstract should include a brief background of the research topic, the problem resolved in the current work, what has been done, as well as future work. Apart from that, lines 53 – 57 are repetitive to what has been stated in lines 38 – 40.
- Unclear difference between the two TEM images in Fig. 2. The authors wrote on lines 240 – 241, “Hr-TEM of nanoparticles after drug loading shows closure of the micro-channels indicating penetration of the drug into the micropores (fig2).” However, it is unclear what is the micro-channel structure the authors are indicating and how to identify the closure of such channels from TEM images. Also, additional TEM images need to be included to show that this is not simply one special case.
- For Hr-SEM images provided in Fig. 3, the two images show different magnification, making it inappropriate to compare the small features. It is hard to know if the features, highlighted by the authors, are nanoparticles or fragments of silicate cement.
Minor comments:
- It is suggested to include a schematic illustrating the preparation of the materials and how the materials are applied for different in vitro tests.
- Abbreviations of experimental techniques listed in line 120 need to be explained.
- Line 165, “37OC.” This error also repeatedly appears throughout the text.
- For Figure 4, legends are too small, and they cannot be resolved.
- For Figure 8, there is no scale bar for the images.
Author Response
|
Sr.no |
Reviewers comment |
Action taken |
Pg. no. |
|
1. |
2 first paragraphs (lines 60-70). Supporting references here are not sufficiently new and recent, which does not allow us to assess how relevant and promising this work is. |
Changed the citations 1 and 4 and updated current information (yellow highlighter) |
Page 2 |
|
2. |
Line 76. The end of the sentence needs supporting reference. Here, please add more information about “a tricalcium silicate-based material”. |
End sentence removed More information on tricalcium silicate based material provided (green highlighter) |
Page 2 |
|
3. |
Line 76-80. This information looks random and other points of view/experiments/approaches are not mentioned or explained/
|
Other points of view explained Experimental approaches mentioned (Green highlighter) |
Page 2 and 3 |
|
4. |
Lines 90-97. This paragraph is also a retelling of obsolete data and new data/achievements are required. |
Current information on bioglass nanoparticles provided in a paragraph (blue highlighter) |
Page 3 |
|
5. |
Line 104-113. Please be more clear what new data for the material science are supposed to be obtained and what will be the fundamental novelty. |
Data for material science provided (purple highlighter) |
Page 3 |
|
6. |
Line 126. Correct degrees Celsius. |
Rectified (yellow highlighter) |
Page 4 |
|
7. |
Correct peak values/units (29O and 32O). Furthermore, these data are in contradiction with Fig.1 (XRD). Quality of both XRD and FTIR pictures are very poor. As for the HrTEM picture, it is not clear what influence the radiation damage had here during the measurement? It is important to note that the outcome of apatite and calcite materials indeed depends on their resistance to aging, including radiation. See: |
The image quality rectified
Contradicting data rectified
An additional FTIR Data of drug loaded nanoparticles added |
Page 7
Page 6
Page 6& page 10 |
|
8. |
4 and 5. The quality of the drawings is unsatisfactory, the signatures are either missing or indistinguishable. |
Figure 4 and 5 images changed |
Page 11, 12 |
|
9. |
Before the conclusion, it is necessary to clearly formulate what new data interesting for the material of science were obtained and compare them in the form of a table with the available analogues. |
New data interesting material science is summarised |
Page 16 |
Reviewer 1 comments
Reviewer 2 comments
|
Sr.no |
Reviewers comment |
Action taken |
Pg. no |
|
1. |
The abstract needs to be further polished and detailed experimental results should not be included in the abstract. The abstract should include a brief background of the research topic, the problem resolved in the current work, what has been done, as well as future work. Apart from that, lines 53 – 57 are repetitive to what has been stated in lines 38 – 40. |
Rectified |
Page 1 and 2
|
|
2. |
Unclear difference between the two TEM images in Fig. 2. The authors wrote on lines 240 – 241, “Hr-TEM of nanoparticles after drug loading shows closure of the micro-channels indicating penetration of the drug into the micropores (fig2).” However, it is unclear what is the micro-channel structure the authors are indicating and how to identify the closure of such channels from TEM images. Also, additional TEM images need to be included to show that this is not simply one special case.
|
Additional TEM images provided
Arrows indicate the microchannels, and increased density of in the nanoparticles seen after drug loading indicating drug loading
|
Page 8, 9 |
|
3. |
For Hr-SEM images provided in Fig. 3, the two images show different magnification, making it inappropriate to compare the small features. It is hard to know if the features, highlighted by the authors, are nanoparticles or fragments of silicate cement.
|
Different images before and after addition of nanoparticles with comparable magnification |
Page 10 |
|
4. |
It is suggested to include a schematic illustrating the preparation of the materials and how the materials are applied for different in vitro tests.
|
Schematics added |
Page 14 |
|
5. |
Abbreviations of experimental techniques listed in line 120 need to be explained.
|
Rectified (yellow highlighter) |
Page 3 |
|
6. |
Line 165, “37OC.” This error also repeatedly appears throughout the text.
|
Rectified |
Page 11 |
|
7. |
For Figure 4, legends are too small, and they cannot be resolved. |
Rectified |
Page 10, 11 |
|
8. |
For Figure 8, there is no scale bar for the images. |
Rectified |
Page 13 |

Reviewer 2 Report
This paper is undoubtedly useful and interesting, but in this form it cannot be recommended for publication yet and some points need to be clarified more clearly.
- 2 first paragraphs (lines 60-70). Supporting references here are not sufficiently new and recent, which does not allow us to assess how relevant and promising this work is.
- Line 76. The end of the sentence needs supporting reference. Here, please add more information about “a tricalcium silicate-based material”.
- Line 76-80. This information looks random and other points of view/experiments/approaches are not mentioned or explained/
- Lines 90-97. This paragraph is also a retelling of obsolete data and new data/achievements are required.
- Line 104-113. Please be more clear what new data for the material science are supposed to be obtained and what will be the fundamental novelty.
- Line 126. Correct degrees Celsius.
- Line 233. Correct peak values/units (29O and 32O). Furthermore, these data are in contradiction with Fig.1 (XRD). Quality of both XRD and FTIR pictures are very poor. As for the HrTEM picture, it is not clear what influence the radiation damage had here during the measurement? It
is important to note that the outcome of apatite and calcite materials indeed depends on their resistance to aging, including radiation. See:
Bystrova, A.; Dekhtyar, Y.D.; Popov, A.; Coutinho, J.; Bystrov, V. Modified hydroxyapatite structure and properties: Modeling and synchrotron data analysis of modified hydroxyapatite structure. Ferroelectrics 2015, 475, 135–147.
Hübner, W.; Blume, A.; Pushnjakova, R.; Dekhtyar, Y.; Hein, H.-J. The influence of X-ray radiation on the mineral/organic matrix interaction of bone tissue: An FT-IR microscopic investigation. Int. J. Artif. Organs 2005, 28, 66–73.
Kabacińska, Z., Krzyminiewski, R., Tadyszak, K., & Coy, E. (2019). Generation of UV-induced radiation defects in calcite. Quaternary Geochronology, 51, 24-42.
- 4 and 5. The quality of the drawings is unsatisfactory, the signatures are either missing or indistinguishable.
- Before the conclusion, it is necessary to clearly formulate what new data interesting for the material of science were obtained and compare them in the form of a table with the available analogues.
Author Response
|
Sr.no |
Reviewers comment |
Action taken |
Pg. no. |
|
1. |
2 first paragraphs (lines 60-70). Supporting references here are not sufficiently new and recent, which does not allow us to assess how relevant and promising this work is. |
Changed the citations 1 and 4 and updated current information (yellow highlighter) |
Page 2 |
|
2. |
Line 76. The end of the sentence needs supporting reference. Here, please add more information about “a tricalcium silicate-based material”. |
End sentence removed More information on tricalcium silicate based material provided (green highlighter) |
Page 2 |
|
3. |
Line 76-80. This information looks random and other points of view/experiments/approaches are not mentioned or explained/
|
Other points of view explained Experimental approaches mentioned (Green highlighter) |
Page 2 and 3 |
|
4. |
Lines 90-97. This paragraph is also a retelling of obsolete data and new data/achievements are required. |
Current information on bioglass nanoparticles provided in a paragraph (blue highlighter) |
Page 3 |
|
5. |
Line 104-113. Please be more clear what new data for the material science are supposed to be obtained and what will be the fundamental novelty. |
Data for material science provided (purple highlighter) |
Page 3 |
|
6. |
Line 126. Correct degrees Celsius. |
Rectified (yellow highlighter) |
Page 4 |
|
7. |
Correct peak values/units (29O and 32O). Furthermore, these data are in contradiction with Fig.1 (XRD). Quality of both XRD and FTIR pictures are very poor. As for the HrTEM picture, it is not clear what influence the radiation damage had here during the measurement? It is important to note that the outcome of apatite and calcite materials indeed depends on their resistance to aging, including radiation. See: |
The image quality rectified
Contradicting data rectified
An additional FTIR Data of drug loaded nanoparticles added |
Page 7
Page 6
Page 6& page 10 |
|
8. |
4 and 5. The quality of the drawings is unsatisfactory, the signatures are either missing or indistinguishable. |
Figure 4 and 5 images changed |
Page 11, 12 |
|
9. |
Before the conclusion, it is necessary to clearly formulate what new data interesting for the material of science were obtained and compare them in the form of a table with the available analogues. |
New data interesting material science is summarised |
Page 16 |
Reviewer 2 comments
Reviewer 1 comments
|
Sr.no |
Reviewers comment |
Action taken |
Pg. no |
|
1. |
The abstract needs to be further polished and detailed experimental results should not be included in the abstract. The abstract should include a brief background of the research topic, the problem resolved in the current work, what has been done, as well as future work. Apart from that, lines 53 – 57 are repetitive to what has been stated in lines 38 – 40. |
Rectified |
Page 1 and 2
|
|
2. |
Unclear difference between the two TEM images in Fig. 2. The authors wrote on lines 240 – 241, “Hr-TEM of nanoparticles after drug loading shows closure of the micro-channels indicating penetration of the drug into the micropores (fig2).” However, it is unclear what is the micro-channel structure the authors are indicating and how to identify the closure of such channels from TEM images. Also, additional TEM images need to be included to show that this is not simply one special case.
|
Additional TEM images provided
Arrows indicate the microchannels, and increased density of in the nanoparticles seen after drug loading indicating drug loading
|
Page 8, 9 |
|
3. |
For Hr-SEM images provided in Fig. 3, the two images show different magnification, making it inappropriate to compare the small features. It is hard to know if the features, highlighted by the authors, are nanoparticles or fragments of silicate cement.
|
Different images before and after addition of nanoparticles with comparable magnification |
Page 10 |
|
4. |
It is suggested to include a schematic illustrating the preparation of the materials and how the materials are applied for different in vitro tests.
|
Schematics added |
Page 14 |
|
5. |
Abbreviations of experimental techniques listed in line 120 need to be explained.
|
Rectified (yellow highlighter) |
Page 3 |
|
6. |
Line 165, “37OC.” This error also repeatedly appears throughout the text.
|
Rectified |
Page 11 |
|
7. |
For Figure 4, legends are too small, and they cannot be resolved. |
Rectified |
Page 10, 11 |
|
8. |
For Figure 8, there is no scale bar for the images. |
Rectified |
Page 13 |

Round 2
Reviewer 1 Report
This reviewer appreciates the authors’ careful consideration of my previous comments and offering of the additional data. However, after evaluating the new data, the questions about high-resolution SEM and TEM images are unsolved. This reviewer suggests the authors consult other researchers familiar with electron micrographs to evaluate if there are convincing structural differences between TEM and SEM images. More importantly, these results may not even be crucial to their applications and they may not need to be included in this manuscript. Below are some detailed comments for the authors if they wish to further modify their manuscripts.
- For TEM images, there are no obvious differences between nanoparticles before and after drug loading, which is predicted. It is extremely difficult to visualize organic molecules directly from electron microscopy unless the authors can find references showing similar results. A potential method to get convincing results is from element mapping such as EELS.
- For SEM images, there are also dot-like features in SEM images before adding the nanoparticles. Thus, it is inappropriate to only label arrows after adding the nanoparticles. Unless the author can offer elemental mapping these nanoparticles are indeed different from the cement materials, the conclusions drawn from SEM images are not convincing.
Reviewer 2 Report
the authors tried to improve the article, but it all looks unconvincing ...